# Humanized NSG Mouse Models as a Preclinical Tool for Translational Research in Inflammatory Bowel Diseases

**DOI:** 10.3390/ijms241512348

**Published:** 2023-08-02

**Authors:** Veronika Weß, Paula Schuster-Winkelmann, Yasemin Hazal Karatekin, Simge Malik, Florian Beigel, Florian Kühn, Roswitha Gropp

**Affiliations:** 1Department of General, Visceral und Transplantation Surgery, Hospital of the Ludwig-Maximilian University Munich, Nussbaumstr. 20, 80336 Munich, Germany; veronika.wess@med.uni-muenchen.de (V.W.); paula.winkelmann@med.uni-muenchen.de (P.S.-W.); hazal.karatekin@campus.lmu.de (Y.H.K.); simge.malik@med.uni-muenchen.de (S.M.); roswitha.gropp@med.uni-muenchen.de (R.G.); 2Department of Medicine II, Hospital of the Ludwig-Maximilian University Munich, Marchioninistr. 15, 81377 München, Germany; florian.beigel@med.uni-muenchen.de

**Keywords:** animal models, mouse models, immune-deficient, IBD, inflammatory bowel disease, Crohn’s disease, ulcerative colitis, NOD/SCID/IL2rγ^null^, NSG

## Abstract

The development of animal models reflecting the pathologies of ulcerative colitis (UC) and Crohn’s disease (CD) remains a major challenge. The NOD/SCID/IL2rγ^null^ (NSG) mouse strain, which is immune-compromised, tolerates the engraftment of human peripheral blood mononuclear cells (PBMC) derived from patients with UC (NSG-UC) or CD (NSG-CD). This offers the opportunity to examine the impact of individual immunological background on the development of pathophysiological manifestations. When challenged with ethanol, NSG-UC mice exhibited a strong pro-inflammatory response, including the development of edemas, influx of human T cells, B cells and monocytes into the mucosa and submucosa, and elevated expression of the inflammatory markers CRP and CCL-7. Fibrotic alterations were characterized by an influx of fibroblasts and a thickening of the muscularis mucosae. In contrast, the development of pathological manifestations in NSG-CD mice developed without challenge and was signified by extensive collagen deposition between the muscularis propria and muscularis mucosae, as observed in the areas of strictures in CD patients. Vimentin-expressing fibroblasts supplanting colonic crypts and elevated expression of HGF and TGFß corroborated the remodeling phenotype. In summary, the NSG-UC and NSG-CD models partially reflect these human diseases and are powerful tools to examine the mechanism underlying the inflammatory processes in UC and CD.

## 1. Introduction

Inflammatory bowel disease (IBD), comprising Crohn’s disease (CD) and ulcerative colitis (UC), is an idiopathic, chronic, inflammatory disorder of the gastrointestinal tract. Its prevalence is increasing worldwide, currently affecting 3.5 million people in Europe and North America [1,2]. The immune-pathogenesis of CD and UC is not fully understood; however, it is presently thought that genetic pre-dispositions, along with environmental factors, contribute to a breach of tolerance against the colonic microbiome, leading to excessive intestinal inflammation [3,4,5]. Currently, the differential diagnosis of UC and CD relies on endoscopic and histological analyses [6]. UC is characterized by superficial mucosal inflammation and rectal bleeding restricted to the colon. In contrast, CD involves transmural inflammation, which can cause perforation, strictures and extensive fibrosis, and can affect the entire gastrointestinal tract. UC and CD are both umbrella diagnoses covering multiple disease forms that are distinguished by clinical manifestation, severity, course and response to therapeutics. As a result, IBD patients frequently require multiple medical treatment approaches during the course of their disease, causing severe side effects for patients and a high financial burden for health care systems [7]. In cases of a therapy-refractory disease, surgery may be required for both CD and UC [8]. Therefore, there is an urgent medical need for IBD patient stratification for individualized and phase-adapted treatment regimen. This approach has to be matched with animal models to validate the assumed response to a given therapeutic in a selected patient group. In most IBD models, colitis-like symptoms are induced by exposure to chemical substances such as dextran sodium sulfate (DSS) [9] or 2,4,6-trinitrobenzenesulfonic acid (TNBS) [10], causing a highly pro-inflammatory response. In another study, inflammatory symptoms were induced by rectal application of 50% ethanol [11]. However, these models neither reflect the heterogeneous symptoms observed in IBD patients nor the pathophysiological mechanisms of a heterogeneous patient population often suffering from these diseases for decades. To overcome these shortcomings, we developed UC and CD models based on NOD/SCID/IL2rγ^null^ (NSG) mice reconstituted with peripheral blood mononuclear cells (PBMCs) from patients with UC (NSG-UC) or CD (NSG-CD). In these models, the immunological background of the donor is partially maintained, and the observed pathological manifestations include inflammation, edema, crypt elongation, tufting, fibrosis and strictures. The NSG-UC model predominantly exhibits a pro-inflammatory response, while the NSG-CD model demonstrates remodeling of the colon, including extensive fibrosis. In the present study, we characterized and compared both models to highlight their different manifestations and potential advantages over conventional IBD models.

## 2. Results

### 2.1. The NSG-UC Mouse Model

#### 2.1.1. Pathological Manifestations

Immune-compromised NSG mice were reconstituted with PBMCs isolated from patients with UC (Figure 1). As controls, NSG mice reconstituted with PBMCs from a healthy donor were included (NSG-non-IBD). The basic patient characteristics and study groups are listed in Table 1.

To induce disease-specific symptoms, the NSG-UC and NSG-non-IBD mice were challenged twice by rectal application of 10% or 50% ethanol on days 7 and 14, respectively. The mice from both groups experienced mild weight loss upon challenge with ethanol; however, severe diarrhea was only observed in the NSG-UC mice. The clinical symptoms were classified according to a clinical score throughout the study (Figure 2). For analysis, only studies with matched groups were selected (Table 1 UC 1-4; for complete data set, see Appendix A). Table 1 shows the basic patient characteristics and groups defined in this animal model study.

The clinical scores in the unchallenged NSG-non-IBD and NSG-UC mice fluctuated around a mean value of 0.6, most likely reflecting variability in weight measurements. Upon challenge with ethanol, the clinical score increased in both groups significantly, indicating that the rectal application of ethanol had an impact on the mice (mean value of 3.2 and 2.6, respectively, for unchallenged versus challenge; NSG-non-IBD: *p* = 0.055; NSG-UC: *p* = 0.01; ANOVA followed by Tukey’s HSD). On day 18, the mice were euthanized, and their colon was removed, subjected to macroscopic inspection, and classified according to a macroscopic score as described in Section 4. Here, a clear difference was observed between the two models. In contrast to the challenged NSG-non-IBD mice, the colons of the ethanol-challenged NSG-UC mice exhibited unformed pellets, dilatation and colon shortening. In general, the macroscopic inspection of the colon corroborated the clinical score (Figure 2).

To further examine the effect of ethanol, the histopathological manifestations were examined using two different staining methods to either visualize the colon architecture (hematoxylin and eosin, HE) or the connective tissue (Sirius Red, SR), (Figure 3). Various pathological phenotypes were observed in the NSG-UC mice, including edema, influx of inflammatory cells into the mucosa and submucosa, crypt elongation, tufting, goblet cell loss and fibrosis. In areas of complete destruction, fibrosis was not as profound as in areas of mild inflammation (Figure 2, NSG-UC histological score = 16 or 7). In contrast, the NSG-non-IBD mice exhibited hardly any signs of inflammation and only mild edema. Occasionally, fibrotic alterations were observed. The sections were classified according to a histological score described in the Section 4. As shown in Figure 3B, the only significant differences were observed when comparing the unchallenged and challenged NSG-UC mice (*p* = 0) or the challenged NSG-non-IBD and NSG-UC mice (*p* = 4 × 10^−5^; ANOVA followed by Tukey’s HSD).

To corroborate the observation that the immunological background of the donor affected the development of symptoms in the NSG-UC mice, a correlation analysis was performed between the patients’ Simple Clinical Colitis Activity Index (SCCAI) scores and the mean values of the histological scores observed in the mice that were reconstituted with PBMCs from the respective donors. As shown in Figure 4, this correlation was significant (Pearson’s product-moment correlation analysis, cor = 0.49, *p*-value = 0.018, CI = 0.11–1).

To validate the observations obtained from the analysis of clinical scores, inflammatory markers associated with the inflammatory processes were analyzed (UC 3-5). Figure 5 provides an example of common inflammatory markers, such as msCCL-7, msCRP and msIL-6, which were analyzed in protein extracts from the distal parts of the colon and using Luminex. All selected markers were significantly elevated upon challenge with ethanol (CCL-7: unchallenged mean = 679.5 ± 702.31 pg/mL, challenged mean = 3382.81 ± 3540 pg/mL, *p* = 0.001; CRP: unchallenged mean = 6232.37 ± 2642 pg/mL, challenged mean = 9481.12 ± 2807 pg/mL, *p* = 0.01 (Student’s *t*-test); IL-6: unchallenged mean = 5.5 ± 2.9 pg/mL, challenged mean = 1415 ± 3872 pg/mL, *p* = 0.02 (Wilcoxon rank-sum test)).

#### 2.1.2. Characterization of Cells Involved in Inflammation

To gain a better understanding of the immunological compartment of the colon in the NSG-UC mice, an immunohistochemical analysis was performed. As we expected that cells of human origin would migrate into the mucosa and submucosa, anti-huCD4, anti-huCD8, anti-huCD19 and anti-huCD14 antibodies were used. As shown in Figure 6, human CD4 and CD8 T cells, human B cells and human monocytes were detected. Interestingly, these cells seemed to concentrate at the tip of the crypt, suggesting that spatially expressed chemokines direct the migration of human leukocytes. Further analysis to characterize the subtypes of cells needs to be performed in the future.

#### 2.1.3. Fibrotic Alterations of the Colon

As shown in Figure 3, fibrosis is an important part of the inflammatory process in the NSG-UC mice. The fibrotic regions of the sections did not exhibit profound influx of human leukocytes but indicated the influx of fibroblasts. Colonic fibroblasts are a heterogeneous population consisting of at least three different cell types, including subepithelial myofibroblasts (vimentin+, CD90+, and αSMA+), myocytes (vimentin+, αSMA+, and desmin+) and pericytes (vimentin+, CD90−, desmin+, and Acta2 low) [12,13,14]. Therefore, the selected sections were stained with anti-vimentin, anti αSMA, anti-desmin and anti-TRPA1, which have been previously identified as a potential therapeutic target of fibroblasts and myofibroblasts [15].

As shown in Figure 7A, αSMA was predominantly expressed in the muscularis propria and in the muscularis mucosae. In the NSG-non-IBD sections, the muscularis mucosae was visible as a thin line, and weak staining was also observed between the crypts in different sections. Staining with desmin superposed αSMA staining, indicating the presence of myocytes in the muscularis mucosae. Upon challenge with ethanol, the muscularis mucosae thickened in the NSG-UC mice, and αSMA-positive fibroblasts squeezed between the crypts, replacing the epithelial cells. As these cells were desmin negative, these fibroblasts were most probably myofibroblasts. This observation was also corroborated by anti-vimentin staining. Vimentin was extremely weak in the NSG-non-IBD sections and intense in areas that had been identified as fibrotic by H&E staining, as shown in Figure 2. Some fibroblasts were double positive for vimentin and TRPA1, suggesting that suppressing TRPA1 may affect fibrosis. To support these findings, fibroblasts were isolated from the colons of the mice (Figure 7B) and cultivated for five days. As expected, the H&E staining displayed a heterogenic morphology of fibroblasts, suggesting a heterogeneous population in the isolates. All cells expressed vimentin, thus confirming the isolation of fibroblasts. However, the expression pattern of other markers reflected the heterogeneity of the fibroblastic population. αSMA, desmin and TRPA1 were not expressed in every cell.

#### 2.1.4. Testing of Therapeutics in the NSG-UC Mouse Model

The NSG-UC mouse model has become a well-accepted model for validating the efficacy of novel and approved therapeutics addressing human molecular targets (Table 2). This provides a significant advantage as it avoids the need to develop murine surrogate inhibitors and allows inhibition of human-specific pathways, which may differ or may not exist in conventional murine models. In this study, we tested eleven different therapeutics. 

One noticeable issue in this study is inherent variability, which includes the impact of the immunological background of the donor and the variability in scores and expressions of markers within each group. Therefore, a comprehensive analysis of all markers is necessary to obtain reliable results. In most of our studies, we applied OPLS-DA analysis, which not only allows a comparison of the efficacy of therapeutics but also provides quantitative data. Figure 8 presents an example of an OPLS-DA analysis between Infliximab- and Tofacitinib-treated mice (Figure 8). Tofacitinib showed lower R2X and higher R2Y and Q2y values, and a higher RMSSE value. 

### 2.2. The NSG-CD Mouse Model

The impact of the immunological background of the donor was also obvious in the NSG-CD model. While the treatment scheme was identical to the NSG-UC model, the pathological manifestations differed (Figure 9) [24]. In the NSG-CD mice, the development of edema and the influx of inflammatory cells were less prominent compared to the NSG-UC mice, and complete destruction of the mucosa was rarely observed. However, a thickening of the muscularis mucosae and an enlargement of the submucosa were frequently observed (Figure 9A). These areas exhibited collagen deposition and the presence of fibroblasts and fibrocytes (Figure 9B). Additionally, unlike the NSG-UC mice, there was no significant difference in the histological score between the unchallenged (mean = 2.21 ± 1.47) and challenged mice (mean = 3.53 ± 2.02), suggesting that patient-derived PBMCs can spontaneously induce remodeling of inflammatory response (Figure 2). In this analysis, only matched groups were selected (CD1, 3). Human CD8+ cytotoxic T cells were detected in the sections of the NSG-CD mice (Figure 9B), although the influx of inflammatory cells was negligible compared to the NSG-UC mice.

The detected levels of inflammatory markers corroborated the observation of a less inflammatory phenotype in the NSG-CD model (Figure 10). Levels of CRP and CCL-7 were significantly lower in the NSG-CD mice (CRP: NSG-UC mean = 8469.01 ± 3420.11 pg/mL, NSG-CD mean = 3671.4 ± 2736.52 pg/mL, *p* = 0; CCL-7: NSG-UC mean = 2724.23 ± 3425.72, NSG-CD mean = 651.08 ± 1349.15, *p* = 0.0035; Welch two-sample *t*-test). Conversely, remodeling markers like HGF and TGFß were elevated in the NSG-CD mice (HGF: NSG-UC mean = 4.66 ± 3.06 ng/mL, NSG-CD mean = 46.48 ± 35.94 ng/mL, *p* = 2 × 10^−5^; TGFß: NSG-UC mean = 26.13 ± 7.01 g/mL, NSG-CD mean = 31.00 ± 10.11, *p* = 0.02). These observations also suggest that the NSG-CD model reflects the human disease.

## 3. Discussion

The gap between preclinical mouse models and human diseases appears to be irreconcilable due to species-specific signalling pathways, non-matching cellular populations and physiological differences in organ function. This disparity is particularly evident in chronic inflammatory diseases, which exhibit heterogeneous manifestations influenced by factors such as age, disease duration, genetic predisposition and individual etiopathologies. Despite these challenges, mice remain the preferred species for practical reasons. Conventional mouse models typically induce pathophysiological symptoms through the application of chemical substances like DSS or TNBS, resulting in a predominantly pro-inflammatory response [9,10]. While these models have been valuable for validating anti-inflammatory therapeutics, their results have limited predictive value for clinical trials or therapeutic responsiveness, and they poorly reflect the fibrotic alterations observed in UC and CD.

The developed NSG-IBD models described in this study offer several advantages over conventional models. First, they provide a more accurate reflection of these human diseases. From an evolutionary point of view, it is reasonable to assume that the inflammatory processes in IBD are aberrant wound healing processes. These processes involve the recruitment of pro-inflammatory T cells, B cells, monocytes and neutrophils to protect the epithelial barrier from invading pathogens. In the NSG-UC model, this response was observed with the development of edema filled with a mixed infiltrate of leukocytes upon ethanol application. Additionally, human T cells, B cells and monocytes migrated to the tip of the crypts for further protection. Similar to UC, these uncontrolled processes led to the destruction of the mucosa. Like in wound healing processes, a second arm of inflammation was observed which involved fibrosis to seal the damaged area. While variations of fibrosis occurred in the NSG-UC mice, fibrotic alterations were more pronounced in the NSG-CD mice at the expense of a pro-inflammatory phenotype. Invading fibroblasts squeezed and replaced the crypts and the CD-NSG mice exhibited profound collagen deposition between the muscularis mucosae and the muscularis propria, resembling the loss of flexibility of the colon seen in strictures of CD. Both models exhibited a coexistence of pro-inflammatory and remodeling processes, with one process often predominating over the other. These observations were supported by the expression of pro-inflammatory markers (CRP and CCL-7) in the NSG-UC mice and remodeling markers (HGF and TGFß) in the NSG-CD mice. 

Fibrotic areas were characterized by collagen deposition and the presence of fibrocytes and fibroblasts. While it is believed that monocyte-derived fibrocytes migrate from the blood to damaged areas [25,26], the origin of inflammatory fibroblasts is less understood. Given the heterogeneity of the mucosal fibroblast population, it is not clear which subtype contributes to the pool of inflammatory fibroblasts. The study by Kinchen et al. [14] suggests that a specific subtype of fibroblasts proliferates in the inflammatory environment of UC, characterized by an elevated expression of chemokine ligand-19, -21, TNF Superfamily Member 14, major histocompatibility invariant chain CD74, clusterin, interleukin-33, CD24 and podoplanin. However, this study does not provide guidance on targeting excessive fibrosis. Another study by West et al. [27] suggests that inflammatory fibrocytes express the oncostatin M receptor, potentially allowing targeting with inhibitors. The origin of these fibroblasts was not elucidated in this study.

Interestingly, fibrosis in the NSG-CD model developed spontaneously without the need for ethanol application. This observation suggests that PBMCs from CD and UC patients carry intrinsic information that leads to the distinct phenotypes in NSG mice. One possible explanation is that an auto-toxic T cell response triggers CD, as a subtype of CD8+ T cells expressing genes indicating a canonical effector phenotype, including KLRG1, GZMB, GZMK, PRF1, IFNG and FCRL6 which have been detected in terminal ileal resections [28]. On the other hand, the response to the mild irritant ethanol in the NSG-UC mice suggests that PBMCs derived from UC donors are differently harnessed and therefore evoke a different pathological phenotype. The specific differences are not yet understood, but these models provide an opportunity to gain a better understanding of the onset and mechanism of UC and CD.

Second, these models partially reflect the inflammatory background of the donor and, when combined with patient profiling, may eventually enable stratification of patients for clinical trials, reducing the risk in clinical studies [23]. Currently, results obtained in animal models indicating efficacy have limited value as 54% of drugs fail in phase II clinical trials due to the lack of efficacy [29]. While inadequate animal models are not the sole cause of failure, the NSG-IBD models offer the possibility of predicting therapeutic response based on the inflammatory profile of patients selected for reconstitution [21,23].

Third, these models allow the testing of therapeutics targeting human molecules, potentially avoiding the need for developing surrogate inhibitors specific to murine targets or testing in non-human primates.

However, there are limitations of the IBD models described in this study. They are still chimeric and reliant on the interaction of murine ligands and cognate human receptors and vice versa. Although certain chemokine and chemokines and chemokine receptors appear to facilitate the directed migration of human leukocytes to the mucosa, the communication between resident murine cells and migrating leukocytes is not fully understood. This may limit the extent to which the NSG-IBD models accurately reflect these human diseases. In addition, these models exhibit inherent variability due to the inflammatory background of the donor, levels of engraftment, and diverse pathological manifestations encompassing both pro-inflammatory responses and various forms of fibrosis. Lastly, they are logistically more complicated than conventional models and require the commitment of patients who are willing to donate blood.

In summary, despite these limitations, we believe that the advantages offered by these models outweigh their drawbacks and that they provide powerful tools to elucidate inflammatory mechanisms in UC and CD.

## 4. Materials and Methods

### 4.1. Isolation of PBMCs and Engraftment

A total of 60 mL of peripheral blood in trisodium citrate solution (S-Monovette, Sarstedt, Nürnberg, Germany) was collected from the arm vein of healthy donors or IBD patients, as described previously [24]. The blood was diluted with Hank’s balanced salt solution (HBSS, Sigma-Aldrich, Deisenhofen, Germany) at a 1:2 ratio. The suspension was loaded onto LeucoSep tubes (Greiner Bio One, Frickenhausen, Germany). PBMCs were separated by centrifugation at 400 g for 30 min without acceleration. The interphase was extracted and diluted with phosphate-buffered saline (PBS) to a final volume of 40 mL. Cells were counted and centrifuged at 1400 g for 5 min. The cell pellet was resuspended in PBS at a concentration of 4 × 10^6^ cells in 100 µL.

### 4.2. Study Protocol

NSG mice were obtained from Charles River Laboratories (Sulzfeld, Germany). The mice were kept under specific pathogen-free conditions in individually ventilated cages in a facility controlled according to the Federation of Laboratory Animal Science Association (FELASA) guidelines. The age of the mice was between 6 and 8 weeks.

Following engraftment on day 1, the mice were presensitized by rectal application of 150 µL of 10% ethanol on day 8 using a 1 mm cat catheter as previously described [23] (Henry Schein, Hamburg, Deutschland). The catheter was lubricated with Xylocain©Gel 2% (AstraZeneca, Wedel, Germany). Rectal application was performed under general anesthesia using 4% isofluran. Post application, the mice were kept at an angle of 30° to avoid ethanol dripping. On day 15, the mice were challenged by rectal application of 150 µL of 50% ethanol following the protocol of day 8. On day 18, the mice were sacrificed. Therapeutic antibodies (150 µL in PBS) were applied i.p. on days 7 and 14, and small-molecule inhibitors (150 µL in PBS or 0.5% methylcellulose gel in PBS (Merck KGaA, Darmstadt, Germany, Firma Cat# M0512) were applied i.p. on days 7–9 and 14–17.

### 4.3. Clinical Activity Score

The assessment of colitis severity was performed daily according to the following scoring system [23]: Loss of body weight: 0% (0), 0–5% (1), 5–10% (2), 10–15% (3), and 15–20% (4). Stool consistency: formed pellet (0), loose stool or unformed pellet (2), and liquid stools (4). Behavior: normal (0), reduced activity (1), apathy (4), and ruffled fur (1). Body posture: intermediately hunched posture (1), and permanently hunched posture (2). The scores were added daily to a total score with a maximum of 12 points per day. Animals who suffered from weight loss > 20%, rectal bleeding, rectal prolapse, self-isolation or a severity score > 7 were euthanized immediately and not taken into count. All scores were added for statistical analysis.

### 4.4. Macroscopic Colon Score

After sacrificing the animals, the colon was removed, a photograph was taken, and the colon was scored for macroscopic characteristics as follows [23]: pellet: formed (0), soft (1), and liquid (2); length of colon: >10 cm (0), 8–10 cm (1), and <8 cm (2); dilation: no (0), minor (1), and severe (2); hyperemia: no (0) and yes (2); and necrosis: no (0) and yes (2).

### 4.5. Histological Analysis

Necropsy samples from the distal parts of the colon were fixed in 4% formaldehyde for 24 h, stored in 70% ethanol and routinely embedded in paraffin. The samples were cut into 3 µm sections and stained with hematoxylin and eosin (H&E) and Sirius Red (SR). 

Epithelial erosions were scored as follows [23]: no lesions (1), focal lesions (2), multifocal lesions (3), and major damage with involvement of basal membrane (4). Inflammation was scored as follows: infiltration of few inflammatory cells into the lamina propria (1), major infiltration of inflammatory cells into the lamina propria (2), confluent infiltration of inflammatory cells into the lamina propria (3), and infiltration of inflammatory cells including tunica muscularis (4). Fibrosis was scored as follows: focal fibrosis (1) and multifocal fibrosis and crypt atrophy (2). The presence of edema, hyperemia and crypt abscess was scored with 1 additional point in each case. The scores for each criterion were added to a total score ranging from 0 to 12. Images were taken with a Zeiss AXIO Observer microscope (Zeiss, Oberkochen, Germany) using the Zeiss ZEN2 lite software. Figures show representative longitudinal sections in original magnification. The SR sections were photographed under polarized light. In Adobe Photoshop CC, a tonal correction was used to enhance contrasts within the pictures.

### 4.6. Detection of Cytokines in Mouse Colon

Approximately 10 mm long sections of the terminal colon were dissected and cleaned of feces with ice-cold PBS. Then, 350 µL of protease inhibitor cocktail (cOmplete, Roche, Penzberg, Germany) was added according to the manufacturer’s instructions. The samples were milled for 5 min at 50 Hz with a 5 mm stainless steel bead (Tissuelyser II, Qiagen, Hilden, Germany), centrifuged for 5 min at 300 g, and 150 µL of supernatants was shock frozen and stored at −80 °C. MsCRP (Thermo Fisher Scientific, Germering, Germany, Cat# EPX01A-26045-901, RRID:AB_2575963 and msTGFß (Thermo Fisher Scientific, Cat# EPX01A-20608-901, RRID:AB_2575921; CCL-7 (Thermo Fisher Scientific Cat# EPX01A-26006-901, RRID:AB_2575933, Darmstadt, Germany) and msHGF was detected via ELISA (Thermofisher Catalog # EMHGF) and determined using Tecan infinite 200 (Mennedorf, Switzerland).

### 4.7. Immunohistochemistry (IHC) and Immune Cytochemistry (ICC)

For the IHC of the colon, the samples from different parts of the murine colon were fixed in 4% formaldehyde for 24 h, stored in 70% ethanol and embedded in paraffin. The samples were cut into 3 µm sections. After de-paraffinization and rehydration with xylene and ethanol, antigen retrieval in 1 mM EDTA was conducted.

For ICC of fibroblast cell cultures, the medium was removed and cells were washed twice with 2 mL of PBS. Following fixation with 2 mL of ROTI^®^Histofix 4% formaldehyde for 15 min at RT, the cells were washed three times with 2 mL of PBS and blocked in 1 mL of blocking buffer (5% FBS in 1 × PBS) at RT for 30 min.

For staining, blocking buffer (1% BSA in PBS) was added for 30 min at room temperature, followed by overnight incubation of the first antibody diluted in 100 µL of blocking buffer at 4 °C and sealed with parafilm (for the antibodies used, see Appendix A). Antibodies were diluted as follows: CD4 1:100, CD8 1:100, CD14 1:100, CD19 1:100, aSMA 1:100, Desmin 1:200, Vimentin 1:100, and TRPA1 1:200. After two washing steps with PBS, the second antibody was added at a concentration of 1:400 in 100 µL of blocking buffer for 1 h at room temperature (Alexa Fluor™ 488 rabbit anti-mouse for Vimentin, α-SMA, CD4, CD14, and CD19; Alexa Fluor™ 647 goat anti-rabbit for TRPA1, CD90, and Desmin). The slides were washed three times with PBS and sealed with cover slides with a mounting medium (anti-fade gold, Thermo Fisher, Darmstadt, Germany).

### 4.8. Ex Vivo Fibroblast Cultivation

Fibroblasts were isolated from NSG mouse colon by adapting the protocol described in [30]. At necropsy, the mouse colon was removed and put on a Petri dish containing ice-cold PBS. The colon was cleaned with ice-cold 1 × PBS with the help of a syringe, cut open lengthwise and transferred into a falcon containing 20 mL of ice-cold 1 × PBS. The ice-cold 1 × PBS was replaced three times until the fluid became clear. The colon was put in a falcon containing 25 mL of ice-cold 1 × HBSS and then incubated for 15 min at 37 °C. The experiment proceeded under a sterile hood afterward. The colon was washed with 1 × HBSS and then transferred into 20 mL of digestion medium (20 mL of complete medium + 20 mg of Dispase II (Sigma Aldrich, Deisenhofen, Germany) + 1.8 mL of 10 mg/mL concentration of Collagenase IV (Sigma Aldrich, Deisenhofen, Germany) in HBSS). The digestion sample was put into a shaking water bath at 37 °C for 1 h 15 min until it looks stringy. During digestion, the samples were vortexed every 15 min. Afterward, the samples were centrifuged at 4 °C at 200 rcf for 5.5 min. The supernatant and remaining solid colon was discarded. The samples were centrifuged at 4 °C at 200 rcf (g) for 5.5 min again. The remaining supernatant was discarded. The pellet was dissolved with ~7.5 mL of complete medium (1 × RPMI 1640 Medium + 10% FBS + 1% Penicillin–Streptomycin). The cells were passed through a 70 μm cell strainer in a fresh tube and 350 µL of cell suspension was seeded on the coverslips (24 × 24 mm) of a 6-well plate. For sterilization, the coverslips were incubated in 70% EtOH for at least 1 h, followed by three washing steps with PBS. A total of 350 μL of complete medium was added into the wells. After incubation at 37 °C and 5% CO_2_ for 3 h, the wells were washed twice with 450 μL of HBSS. Then, 1200 μL of complete medium was added for incubation. After 24 h, the medium was replaced by the same amount of fresh complete medium. The cells could be cultivated up to 7 days. For H&E staining, the cells were permeabilized with 2 mL of 0.5% Tween-20 in PBS for 20 min.

### 4.9. Statistical Analysis

Statistical analysis was performed using R: A language and environment for statistical computing (R Foundation for Statistical Computing, Vienna, Austria. URL https://www.R-project.org/, version 3.6.3, accessed on 29 February 2020). The data were analyzed using Cumming plots [31] using the dabestr package for data presentation and comparison. Cumming plots are a new generation of data analysis with bootstrap-coupled estimation (DABEST) plots that move beyond *p*-values. These plots can be used to visualize large samples and multiple groups easily. For correlation analysis, Pearson’s product-moment correlation was performed and a 95% confidence interval was applied. The variables were also represented as mean, standard deviation and median values. A two-sided Student’s *t*-test with a confidence level = 0.95 was used to compare binary groups, and for more than two groups, ANOVA followed by Tukey’s HSD was conducted. In cases where data were not equally distributed, a Wilcoxon rank-sum test was performed. Orthogonal partial least squares discrimination analysis (oPLS-DA) was performed using the ropls package.

## 5. Conclusions

There is an urgent medical need for accurate IBD patient stratification to optimize patient care and response to therapy. This approach needs to be matched with animal models to validate the expected response to a given therapeutic in a selected patient group. Currently, none of the animal models for IBD fully replicates all the manifestations of the heterogeneous patient groups observed in these human diseases. However, the NSG-UC and NSG-CD models presented in this study significantly narrow the gap between these human diseases and preclinical studies as both models are highly reflective of the respective human diseases. In both models, the disease pattern is dependent on the individual immunological background of the donor. Therefore, combining patient profiling and testing of therapeutics in these models may lead to stratification of patients for individualized and phase-dependent treatment. In addition, it may reduce the risk of novel therapeutics failing in clinical phase II studies.

## Figures and Tables

**Figure 1 ijms-24-12348-f001:**
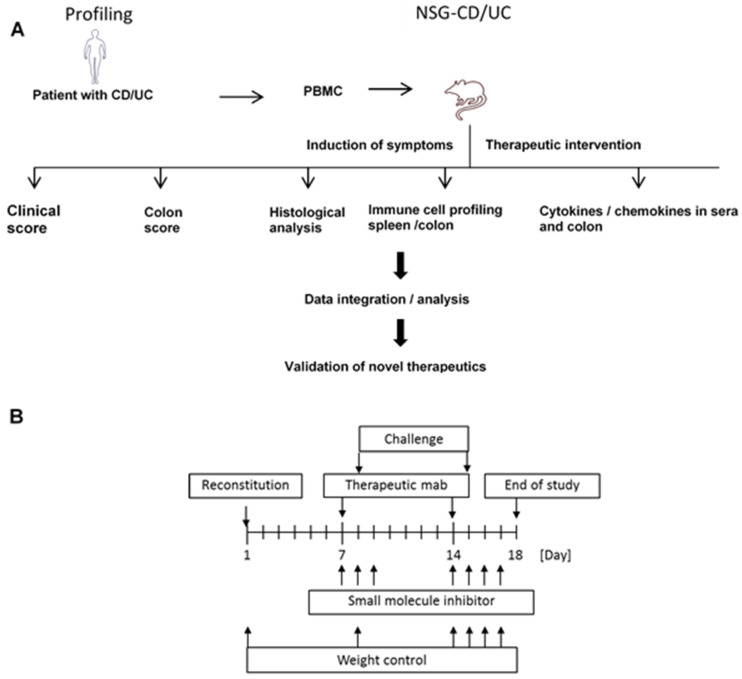
Schematic depiction of the animal model study: (**A**) flowchart and (**B**) scheme. NSG mice were reconstituted through injection of 4 × 10^6^ PBMCs in 100 µL of PBS into the tail vein on day 1 and were left unchallenged or challenged by rectal application of 150 µL of 10% ethanol on day 8, and 150 µL of 50% ethanol on day 15. Therapeutic antibodies (150 µL in PBS) were applied intraperitoneally (i.p.) on days 7 and 14 and small-molecule inhibitors were dissolved in 150 µL of PBS or 0.5% methylcellulose and injected via i.p. on days 7–9 and 14–17.

**Figure 2 ijms-24-12348-f002:**
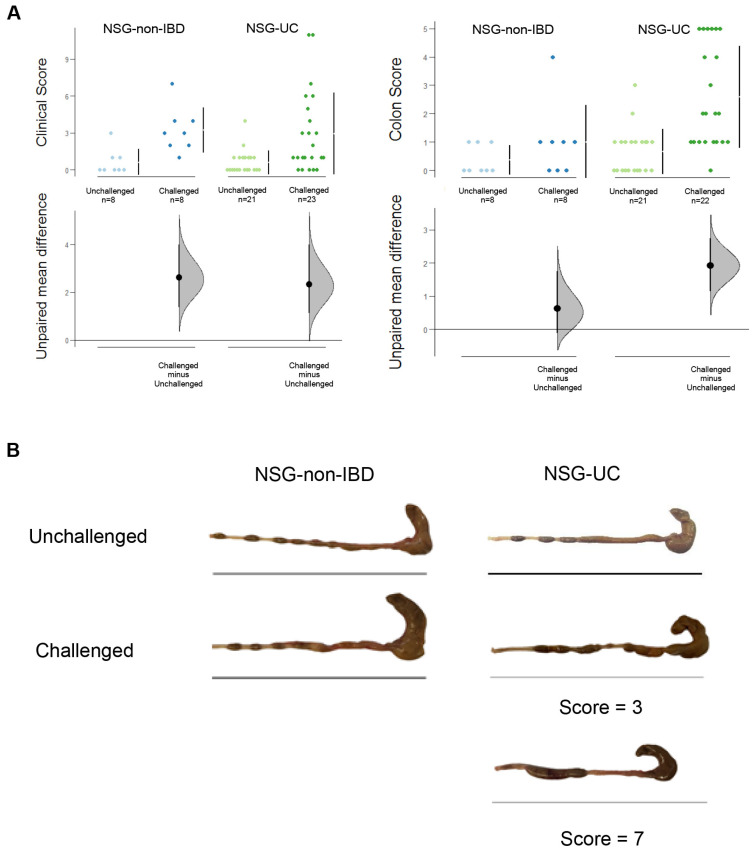
An inflammatory background of the donor and a challenge with ethanol are required to induce symptoms of UC in NSG-UC mice. (**A**) Clinical and macroscopic colon scores of NSG-non-IBD mice (unchallenged N = 2, n = 8; challenged N = 2, n = 8) and NSG-UC mice (unchallenged N = 4, n = 21; challenged N = 4, n = 23) depicted in Cumming plots. NSG mice were reconstituted with PBMCs on day 1 and were left unchallenged or challenged with 10% ethanol on day 8, and 50% ethanol on day 15. The upper part of the plot presents each data point in a swarmplot. The mean and standard deviation (SD) of each group are plotted as a gapped line, where the vertical lines correspond to the mean ± SD and the mean itself is depicted as a gap in the line. The 0 point of the difference axis is based on the mean of the reference group (control). The dots show the difference between groups (effect size/mean difference). The shaded curve shows the entire distribution of excepted sampling error for the difference between the means (the higher the peak, the smaller the error). The error bar in the filled circles indicates the 95% confidence interval (bootstrapped) for the difference between means. N: no. of donors, n: no. of mice. (**B**) Macrophotographs of representative colons. For the challenged NSG-UC mice, colons with a score of 3 and 7 are depicted. The bar indicates 10 cm.

**Figure 3 ijms-24-12348-f003:**
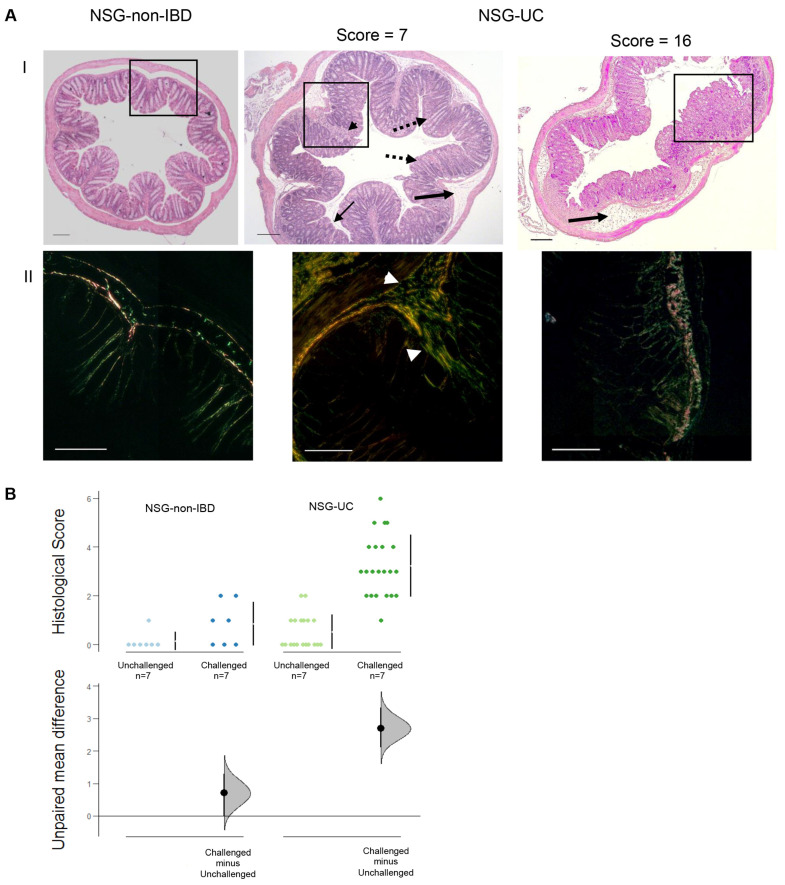
An inflammatory background of the donor and a challenge with ethanol are required to induce pathological manifestations of UC in NSG-UC mice. The mice were treated as described in Figure 2. (**A**) Representative microphotographs of stained sections of distal parts of the colons: (I) H&E staining and (II) SR staining (boxed area). Thick arrows indicate edema and influx of inflammatory cells, dotted lines indicate crypt elongation, thin arrows indicate tufting, and arrow heads indicate fibrosis. Two different sections with different histological scores are depicted and compared to sections from NSG-non-IBD mice. Scale bar represents 100 µm. (**B**) Histological scores depicted as Cumming plot. The upper part of the plot presents each data point in a swarmplot. The mean and standard deviation (SD) of each group are plotted as a gapped line, where the vertical lines correspond to the mean ± SD and the mean itself is depicted as a gap in the line. The 0 point of the difference axis is based on the mean of the reference group (control). The dots show the difference between groups (effect size/mean difference). The shaded curve shows the entire distribution of excepted sampling error for the difference between the means (the higher the peak, the smaller the error). The error bar in the filled circles indicates the 95% confidence interval (bootstrapped) for the difference between means. N: no. of donors, n: no. of mice.

**Figure 4 ijms-24-12348-f004:**
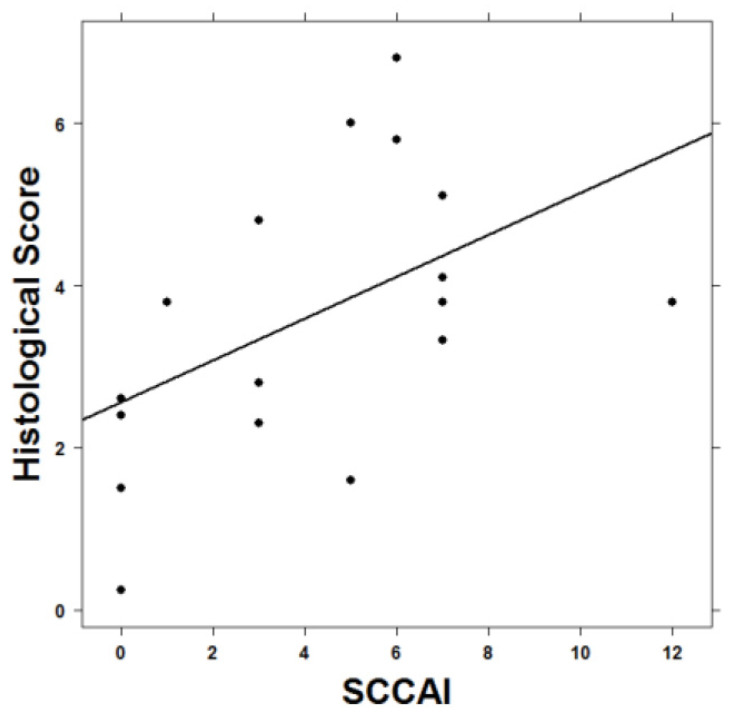
Pearson’s product-moment correlation analysis of murine histological scores and human clinical SCCAI scores depicted as a scatter plot. The SCCAI scores of patients whose PBMCs were used for reconstitution were correlated with the mean values of the histological scores of the respective NSG−UC mice challenged with ethanol.

**Figure 5 ijms-24-12348-f005:**
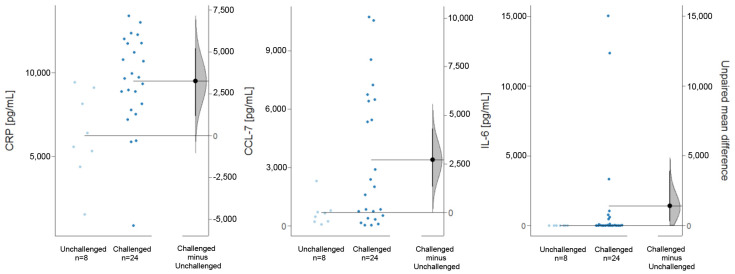
Challenge with ethanol induces the expression of inflammatory markers in NSG-UC mice. The mice were treated as described in Figure 2 (unchallenged N = 2, n = 8; challenged N = 4, n = 24). Levels of msCRP, msCCL−7 and msIL−6 were detected using Luminex from colon extracts and depicted as Cumming plots. The upper part of the plot presents each data point in a swarmplot. The mean and standard deviation (SD) of each group are plotted as a gapped line, where the vertical lines correspond to the mean ± SD and the mean itself is depicted as a gap in the line. The 0 point of the difference axis is based on the mean of the reference group (control). The dots show the difference between groups (effect size/mean difference). The shaded curve shows the entire distribution of excepted sampling error for the difference between the means (the higher the peak, the smaller the error). The error bar in the filled circles indicates the 95% confidence interval (bootstrapped) for the difference between means. The labeling on the right side applies to all graphs. N: no. of donors, n: no. of mice.

**Figure 6 ijms-24-12348-f006:**
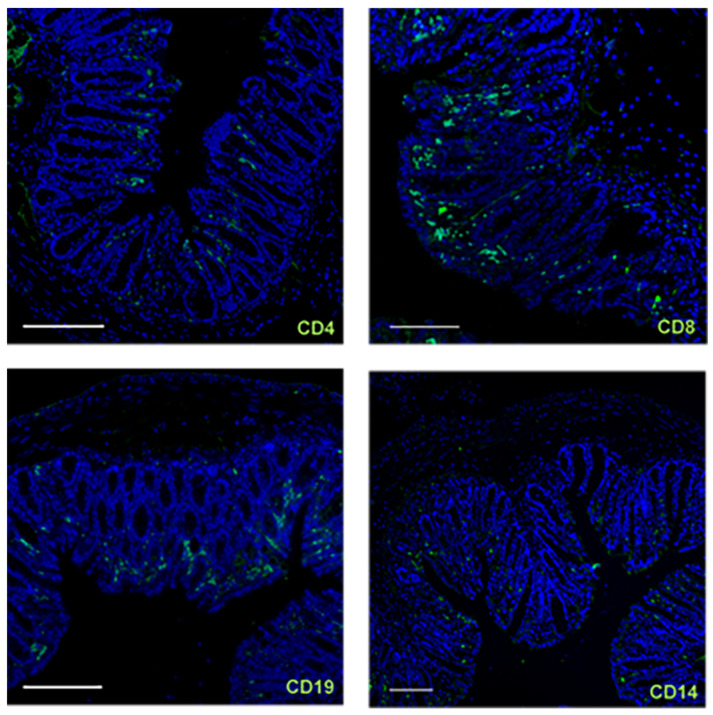
Cells of human origin migrate into the mucosa and submucosa. Sections from the distal parts of the colon were stained with green Alexafluor anti-human CD4, CD8, CD19 and CD14. Scale bar represents 100 µm.

**Figure 7 ijms-24-12348-f007:**
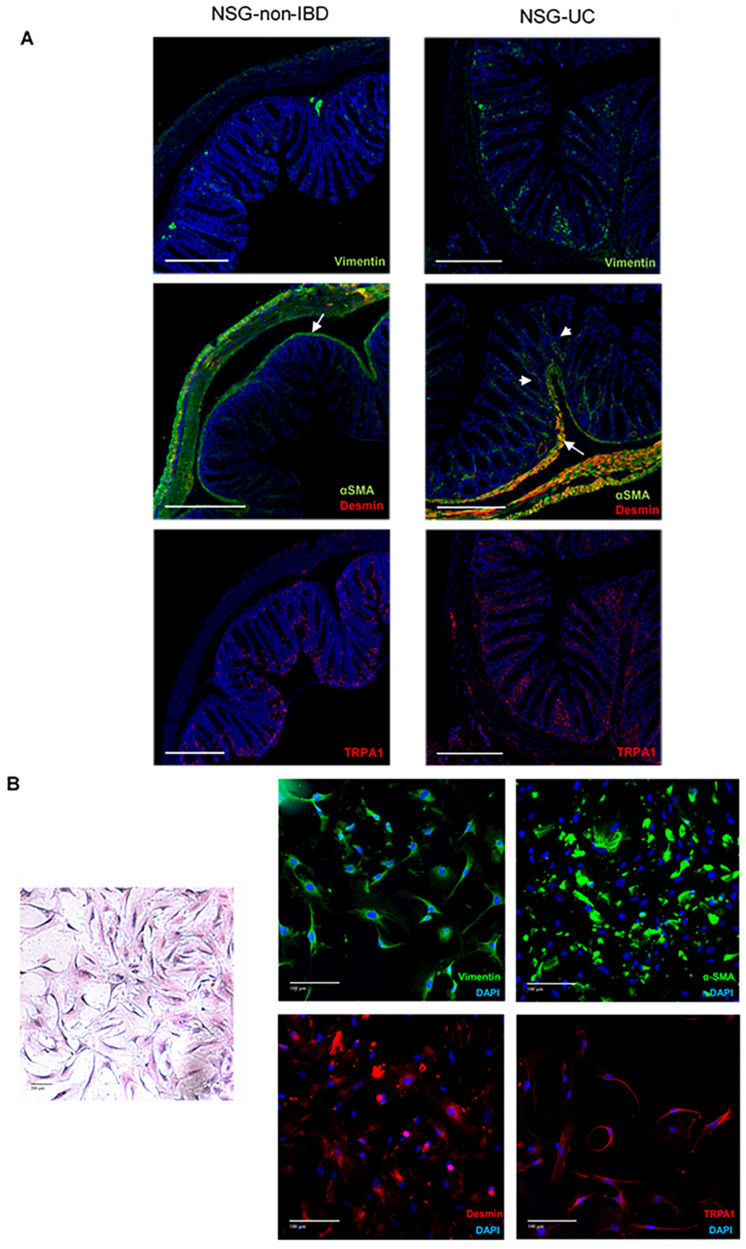
Colonic fibroblasts exhibit a heterogeneous morphology. Immunohistochemical analysis of (**A**) sections of NSG-non-IBD and NSG-UC mice challenged with ethanol and of (**B**) isolated fibroblasts from mouse colons. Arrows indicate muscularis mucosae, and arrow heads indicate influx of myofibroblasts into fibrotic areas. The sections were stained with anti-vimentin, anti-αSMA (green Alexafluor), anti-desmin, anti-TRPA1 (red Alexafluor), DAPI blue and H&E. Scale bar represents 100 µm or 200 µm.

**Figure 8 ijms-24-12348-f008:**
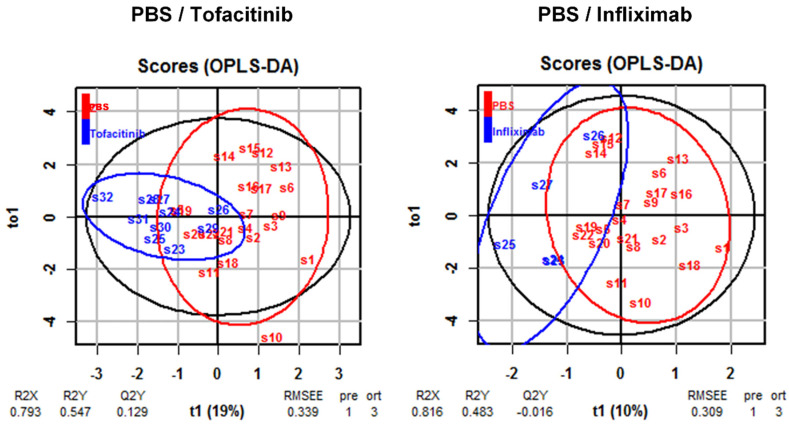
Examples of OPLS-DA analysis of inflammatory parameters obtained from NSG-UC mice treated with Infliximab or Tofacitinib. The mice were treated as described in [21]. Clinical, colonic and histological scores, and levels of msIL−6, msTGFß, msCRP, msCCL−7 and tryptophan were used for modeling. R^2^X: fraction of variation in the × variables explained by the model; R^2^Y: fraction of variation in the Y variables explained by the model, Q^2^Y: fraction of variation in the Y variables predicted by the model; RMSEE: root mean square error of estimation.

**Figure 9 ijms-24-12348-f009:**
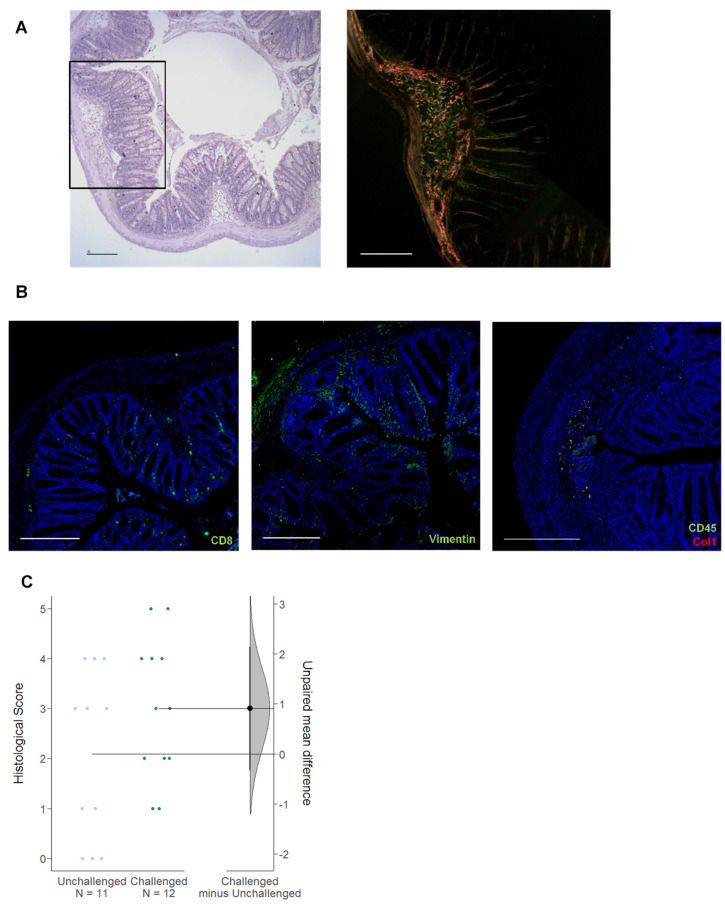
The pathophysiological phenotype in NSG-CD mice. The mice were reconstituted with PBMCs derived from CD donors and treated as described in Figure 2. Sections from the distal parts of the colon of the NSG-CD mice challenged with ethanol were analyzed. (**A**) H&E and SR staining (boxed area). (**B**) Immunohistochemical analysis using anti−human CD45, anti−human CD8, antimurine vimentin (green Alexafluor), and anti−Col1(red Alexafluor). Scale bar represents 100 µm. (**C**) Clinical, colon, and histological scores of challenged and unchallenged NSG-CD mice depicted as Cumming plot. The upper part of the plot presents each data point in a swarmplot. The mean and standard deviation (SD) of each group are plotted as a gapped line, where the vertical lines correspond to the mean ± SD and the mean itself is depicted as a gap in the line. The 0 point of the difference axis is based on the mean of the reference group (control). The dots show the difference between groups (effect size/mean difference). The shaded curve shows the entire distribution of excepted sampling error for the difference between the means (the higher the peak, the smaller the error). The error bar in the filled circles indicates the 95% confidence interval (bootstrapped) for the difference between means. N: no. of donors, n: no. of mice.

**Figure 10 ijms-24-12348-f010:**
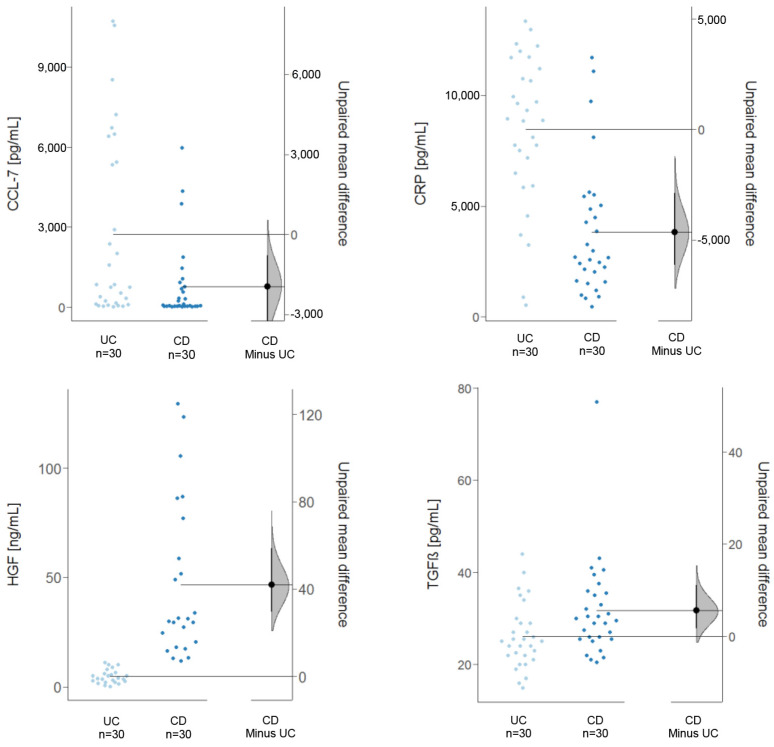
Inflammatory and remodeling markers are differentially expressed in the NSG−UC and NSG−CD models. Levels of CCL−7 (UC: N = 5, n = 30; CD: N = 5, n = 30), CRP (UC: N = 5, n = 30; CD: N = 5, n = 30), TGFß (UC: N = 5, n = 30; CD: N = 5, n = 30), and HGF (UC: N = 4, n = 30; CD: N = 4, n = 24) depicted as Cumming plots. The upper part of the plot presents each data point in a swarmplot. The mean and standard deviation (SD) of each group are plotted as a gapped line, where the vertical lines correspond to the mean ± SD and the mean itself is depicted as a gap in the line. The 0 point of the difference axis is based on the mean of the reference group (control). The dots show the difference between groups (effect size/mean difference). The shaded curve shows the entire distribution of excepted sampling error for the difference between the means (the higher the peak, the smaller the error). The error bar in the filled circles indicates the 95% confidence interval (bootstrapped) for the difference between means. N: no. of donors, n: no. of mice.

**Table 1 ijms-24-12348-t001:** Basic patient characteristics and groups as defined in this animal model study.

Donor	Diagnosis	Age/Gender (f/m)	Medication	Montreal Classification	SCCAI/CDAI	Location				Stricturing Complications	Fibrosis	Groups in the NSG Model	
						Terminal Ileum	Colon	Ileocolon	Upper GI			Control n (f/m)	Ethanol n (f/m)
UC 1	UC	80/f	Infliximab		5							5 (0/5)	5 (2/3)
UC 2	UC	54/m	Vedolizumab, Mesalazine		3							4 (2/2)	6 (0/6); †1
UC 3	UC	30/m	Vedolizumab, Glucocorticoids		7							6 (0/6)	6 (0/6)
UC4	UC	33/f	Vedolizumab, Glucocorticoids, Mesalazine		7							6 (4/2)	6 (0/6)
UC5	UC	34/f	Vedolizumab, Mesalazine		3								6 (0/6)
UC6	UC	32/f	Vedolizumab, Mesalazine		7								7 (5/2)
UC7	UC	24/m	Infliximab		12								5 (0/5)
Mean		41 ± 19.55 (3/3)											
Sum												21 (6/15)	41 (7/34)
CD 1	CD	36/m	Vedolizumab	A2/L3/B2	0	yes	no	yes	no	yes	no	6 (6/0)	6 (4/2)
CD 2	CD	42/w	None	nd	0								6 (5/1)
CD 3	CD	26/m	Filgotinib, Glucocorticoids	A2/L3/B1	0	yes	yes	yes	no	no	no		6 (4/2)
CD 4	CD	26/w	Vedolizumab, Ustekinumab, Glucocorticoids	A2/L4/B2	200	yes	yes	yes	yes	yes	yes	6 (2/4)	6 (4/2)
CD 5	CD	74/m	Glucocorticoids	A2/L3/B2	30	yes	yes	yes	no	yes	yes		6 (4/2)
Mean		40.8 ± 19.77 (2/3)											
Sum												12 (8/4)	30 (21/9)
Non-IBD 1	Non-IBD	24/w										4 (0/4)	4 (4/0)
Non-IBD 2	Non-IBD	65/w										4 (4/0)	4 (0/4)
Mean		44 ± 28.28											
Sum												8 (4/4)	8 (4/4)

† Euthanasia due to a critical score post-challenge.

**Table 2 ijms-24-12348-t002:** Therapeutics tested in the NSG-UC mouse model.

Therapeutic	Target	Outcome	Reference
Anti-CD1a mab	CD1a	ameliorating	[16]
Sirolimus	AKT/mTor	ameliorating	[17]
Pitrakinra	IL-4/IL-13 receptor	deteriorating	[18]
Copanlisib	PI3K	ameliorating	[19]
Ritonavir	GLUT	ameliorating	[20]
DES1	Kv1.3	ameliorating	[21]
Tofacitinib	Jak	ameliorating	[21]
Infliximab	TNFα	ameliorating	[21]
Adalimumab	TNFα	ameliorating	[22,23]
Oxelumab	OX40L	ameliorating	[22]
Anti-CCR4 mab	CCR4	deteriorating	[18]

## Data Availability

All data are provided in the manuscript.

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
