# Peer review of "Humanized NSG Mouse Models as a Preclinical Tool for Translational Research in Inflammatory Bowel Diseases"

_ijms, 2023, doi:10.3390/ijms241512348_

Round 1
Reviewer 1 Report
In this paper, the authors characterized basic aspects of two humanized mouse models of inflammatory bowel diseases, namely NSG-UC and NSG-CD. The NSG-UC model was first described in their publication in 2017, and the NSG-CD model is new. The results are largely descriptive, and the numerical results require some improvement on statistical analyses.
Major points:
1. In lines 230-234, the authors claim that the higher R2x, Q2y, and lower RMSSE value suggested superiority of tofacitinib over infliximab. However, the R2x value for tofacitinib shown in Figure 8 is lower than that of infliximab. The RMSSE is higher for tofacitinib. Moreover, these statistical values are aimed to assess the fitness of the model they used. Their model is build using clinical-, colon-, and histological scores and IL-6, TGF-β, MCP-3, and tryptophan. Because slight differences may be reverted by the choices of the explaining parameters, it is safe not to discuss this superiority using these results. Otherwise, a reference mathematically explaining the statistical significance of the parameter differences is required.
2. In figures 2A, 3B, 5, 9 and 10, Unpaired mean difference graphs are shown. These are not useful where the original dots are depicted, and even misleading in some cases where the distribution does not obey normal distribution.
3. In figures 2A, 3B, AVOVA followed by Tukey HSD is used for analyses. However, the distribution of the data seems not the same among groups. Use other statistical methods unless you show the distribution is following the normal distribution with the same standard deviation.
Minor points:
1. In line 77, what is “a standard protocol”? Is there any experiment where a non-standard protocol used?
2. In line 79, authors describe “severe diarrhea was only observed in NSG-UC mice”. However, there is no data shown. “(Data not shown)” may be required.
3. In Figures 2, 3, 5 and 10, please explain “N” and “n”. I assume N means number of donors while n means the number of mice. In addition, I assume the “N” in figure panel should be “n”.
4. In line 141, abbreviation “SCCAI” is firstly used. Please spell it out.
5. In Figure 4, the statistical values are overwrapping the data dots. Please remove it since the same values are described in the main text.
6. In the paragraph starting from the line 151, the inflammatory markers are described. Are they mouse or human? Please clarify.
7. In Figure 5, why only NSG-UC mice are shown? Explain.
8. In Figure 6 title, the authors claim that the cells migrating the mucosa and submucosa are human origin. It implies that all or most of the cells are human origin. It would be more logical to say that “cells of human origin are migrating into the mucosa and submucosa”.
9. In Figure 6 and 7, how many samples are assessed?
10. In line203, the authors describe the vimentin staining in the area of fibrosis identified in H&E staining. However, H&E staining data is missing.
11. In line 204, the authors claim that the existence of fibroblasts positive for both vimentin and TRPA1 indicates that TRPA1 may affect fibrosis. This is just a speculation. “Indicating” is too strong.
12. The sentence starting line 220 requires references.
1. In line 74, “As control” should be “As controls”.
2. In Table 1. , Column Fibrosis, row CD1, “nein” should be “no”.
3. In line 85, “clinical score” should be “clinical scores”.
Reviewer 2 Report
The manuscript by Veronika Weß et al. describes an exciting advance in animal modeling of IBD.
I have just a few minor points to make.
1. The authors should provide references for the statement "To induce disease-specific symptoms, NSG-UC and NSG-non-IBD mice were challenged twice according to a standard protocol by rectal application of 10% or 50% ethanol on days 7 and 14, respectively." Preferably the cited references should compare ethanol-induced colitis with other models of induced colitis. If the authors think that it is appropriate, one of the references could be the initial description of ethanol-induced colitis by Andrade, Vaz, and Faria.
2. To be consistent with current nomenclature, the authors should use CCL-7 rather than MCP-3.
3. In the last paragraph, the authors write "In addition, these models have an inherent variability due to the inflammatory background of the donor, levels of engraftment, and diverse pathological manifestations to include pro-inflammatory responses and various manifestations of fibrosis." It appears that this is considered a limitation of the model. However, it may actually be a strength of the model as it reflects the human condition. For example, as noted on line 354 "Second, these models partially reflect the inflammatory background of the donor". So, possibly the primary limitation of the model is variation in the levels of engraftment, while variability due to the inflammatory background of the donor and diverse pathological manifestations that include pro-inflammatory responses and various manifestations of fibrosis, are strengths of the model.
The manuscript is easy to read and understand, however, there are numerous errors in English grammar and diction. In addition, there are some typos in the text. For example the sentence on lines 312-314 "These processes include the protection or the epithelial barrier from invading pathogens by attracting pro-inflammatory T-cells, B-cells, monocytes, and neutrophils" should probably be "These processes include the protection of the epithelial barrier from invading pathogens by attracting pro-inflammatory T-cells, B-cells, monocytes, and neutrophils."
Reviewer 3 Report
Dear authors,
After reviewing your study, I am confused to the overall rationale of this study. At 2021, the authors had reported that reconstituting PBMCs from UBD patients into NSG mice can reproduce the pathological phenotype of IBD (10.1002/iid3.516). Therefore, the feasibility of humanized-NSG-IBD model is undoubted. However, in this study, the authors still spent more than half of words in demonstrating the feasibility of the mice model in representing pathological characteristics in IBD rather than testing the curative methods. This strategy does not match the title of this study. In my opinion, despite this study being quite interesting, the structure of the study needs to be remodeled.
By the way, I am curious of two questions:
1. How long can the manifestation retain after ethanol challenge? Additionally, how long can the reconstituted PBMC retain? IBD-associated colorectal cancer is also a critical issue and DSS challenging model is still applied in the pathogenesis of IBD-associated CRC. This model might be suitable for studying IBD-associated CRC as the model can be retained for a long time.
2. Based on the clinical manifestation, course, and underlying mechanism, IBD, both UC and CD, can be subgrouped into several subtypes (10.1136/gutjnl-2016-312518, 10.1126/sciimmunol.abb4432). Also, the immunoprofiling of each IBD patient is highly variant (10.1126/sciimmunol.abb4432). How do the authors ensure that the built model is universal for a particular subtype of IBD rather than the patient only?
Reviewer 4 Report
Dear authors,
The purpose of your research was to demonstrate that NSG-UC and NSG-CD models are highly reflective of the respective human disease. These models, by partially mirroring the inflammatory background of the donors, offer valuable insights into studying the mechanisms underlying inflammatory processes in ulcerative colitis (UC) and Crohn's disease (CD). Additionally, they have the potential to minimize risks in clinical studies.
However, there are several questions that make this article difficult to follow, and need clarification:
Line 52 - “In most IBD models, colitis-like symptoms are induced by exposure toxins such as dextran sodium sulfate (DSS) [9] or 2,4,6-trinitrobenzenesulfonic acid (TNBS), causing a highly pro-inflammatory response.”
For some authors, “toxins” are poisonous substances produced by living cells or organisms, but others refer to toxins as they would any poison and call those toxins that have a living source ‘biotoxins’ or ‘natural toxins’. To avoid this confusion, it is necessary to clarify that DSS or TNBS are chemicals:
*DSS is a sulfated polysaccharide with variable molecular weights. DSS colitis is the result of a disruption in epithelial barrier.
*TNBS is a nitroaryl oxidizing acid that functionates as an hapten – it binds to tissue protein turns into an antigen and elicits number of immunologic responses. TNBS-induced colitis is a delayed-type hypersensitivity reaction to haptenized proteins.
* Table 1 – a) Were equally represented men and women?
b) What was the patients and the donors age average?
c) CD1 fibrosis and CD5 Upper GI – its is written “nein” instead of “no”
Figure 2: The image quality makes it difficult to read the legend in part A
Figure 3: Can you explain why, in part A, why a colon with a score=7 has more fibrosis than another with a score=16??
* Therapeutics tested in the NSG-UC mouse model (table 2) used this same protocol?
For how long were the drugs administered, and how long after the sensitization?
* Figure 9: should include clinical score and colon score of NSG-CD mice, like in figure 3
* Regarding animal protocol, could you detail:
a) total number os animals used
b) number of male and female, and their age
c) Detail engraftment procedure: administration route; injected volume; time between engraftment and pre-sensitization. It would help if you included that information on figure 1.
d) mortality rate (engraftment procedure plus sensitization procedure). Were all deaths due to euthanasia?
Line 424 - “At autopsy, samples from distal parts of the colon…”. An autopsy is a postmortem examination performed on a human. A necropsy is the appropriate term used for any such evaluation performed on an animal. Again 473 .
Thank you for presenting your work and I hope these suggestions are helpful.
I wish you luck and success.
Table 1: CD1 fibrosis and CD5 Upper GI – its is written “nein” instead of “no”
Author Response
please see attchment

Round 2
Reviewer 3 Report
Dear authors,
Thank you for your kindly response to my question. Your response generally eliminate my concern except one. As being a translational model, testing whether the model can reproduce the clinical manifestation from the same treatment is also important. Do the authors have any idea about the clinical response of the treatment identical to the donor?
Author Response
Reviewer's point:
Thank you for your kindly response to my question. Your response generally eliminate my concern except one. As being a translational model, testing whether the model can reproduce the clinical manifestation from the same treatment is also important. Do the authors have any idea about the clinical response of the treatment identical to the donor?
Response:
We agree that a study like that would be the ultimate proof of concept for the model. So far, we did not have the resources but we plan to perform a study with patients who are either naïve or switched to a different therapy and reconstitute mice simultaneously to treat them like the respective donors.
Reviewer 4 Report
Dear authors,
I appreciate the fact that you have addressed the raised issues and made the necessary changes, resulting in an overall improvement of the manuscript.
Please review Table 1, CD1 line.
Thank you for resubmitting your work.
I wish you luck and success.
Please review Table 1, CD1 line: 36/"M" and "nein"
Author Response
Reviewer's point:
I appreciate the fact that you have addressed the raised issues and made the necessary changes, resulting in an overall improvement of the manuscript.
Please review Table 1, CD1 line.
Response:
We are sorry for having overlooked this point. Thank you for noticing.